# Surgical Site Infection After Posterior Spinal Fusion for Paediatric Spinal Deformities: A Single-Centre Retrospective Observational Study

**DOI:** 10.3390/healthcare13233043

**Published:** 2025-11-25

**Authors:** Dimitrios P. Christakos, Ioannis S. Benetos, Elias Vasiliadis, Panagiotis Karampinas, Angelos Kaspiris, Patra Koletsi, Ioanna Paspati, Spyridon G. Pneumaticos

**Affiliations:** 13rd Department of Orthopaedic Surgery, National Kapodistrian University of Athens, KAT Hospital, 14561 Athens, Greece; ibenetos@med.uoa.gr (I.S.B.); karapana@yahoo.com (P.K.); angkaspiris@hotmail.com (A.K.);; 2Pediatric Intensive Care Unit, Penteli Children’s Hospital, 15236 Athens, Greece; pkoletsi@gmail.com; 3Department of Orthopaedics, Penteli Children’s Hospital, 15236 Athens, Greece; ipaspati@yahoo.com

**Keywords:** PSF, adolescent, spinal deformity, SSI

## Abstract

Background/Objectives: Surgical Site Infections (SSIs) are among the most common complications of Posterior Spinal Fusion (PSF) in children and adolescents. The rate of SSIs after PSF varies from 0.9% to 3% for idiopathic scoliosis and can be as high as 8.7% for neuromuscular scoliosis due to cerebral palsy. Major factors associated with SSIs include patient’s underlying pathology and comorbidities, the complexity of the procedure, and many extrinsic factors such as the expertise of the surgeon, perioperative antibiotic prophylaxis, length of hospitalisation, and perhaps environmental factors in the operating theatre and the hospital infections rates of the centre at which the procedure is being performed. We sought to identify the overall rate of SSI and possible modifiable risk factors for SSI in children and adolescents treated with PSF in Penteli Children’s Hospital. Methods: A total of 46 consecutive patients accounting for 67 surgeries performed between 2019 and 2024 were included in this retrospective observational study. Inclusion criteria were as follows: patient’s age at the time of surgery less than 22 years, patients treated with PSF only, and more than 9 months postoperative observation. SSI was defined as infection occurring within 90 days of the index procedure. Results: The rate of deep SSI in our department was 6.0%. Older age at the time of surgery and a history of previous spine surgery were risk factors for developing an SSI. Conclusions: Between 2019 and 2024 four cases of deep SSI occurred in our institution, leading to a rate of 6.0% among all PSF cases for this specific time period. Higher age and a history of previous spine surgery were risk factors for SSI in this cohort of patients.

## 1. Introduction

Paediatric spinal deformity is a condition that significantly affects a child or an adolescent in many aspects of everyday life. It causes pain, limitations in physical activity, and significant psychological burden. When conservative methods fail to treat such conditions, surgical treatment is necessary.

Spinal fusion for paediatric deformities is a major operation that is associated with a high risk of complications [1]. In a retrospective review of 19,360 cases from the Scoliosis Research Society M&M Database, rates of complications ranged from 6.3% for idiopathic scoliosis to 17.9% for neuromuscular scoliosis [2]. In another study of 2005 patients undergoing anterior, posterior, or combined spinal fusion, Pugely et al. reported that 10% of patients had a complication in the first month post-surgery and that Anterior Spinal Fusion (ASF) when compared with Posterior Spinal Fusion (PSF) was associated with higher complications rates (28.6% vs. 9.1%) [3].

Surgical Site Infection (SSI) is one of the well-documented complications, which is associated with substantial medical, emotional, and financial burdens. SSI rates after spinal deformity surgery range from 0.5% to 6.7% for Adolescent Idiopathic Scoliosis (AIS), from 6.1% to 15.2% for scoliosis due to cerebral palsy, and from 8% to 41.7% for scoliosis in patients with myelodysplasia. SSIs can lead to overexposure to antibiotics and antimicrobial resistance, increased length of hospital stay, one or more additional surgeries, implant removal, and loss of achieved deformity correction, all of which have a serious socioeconomic burden [4,5].

In 2024, a series of SSIs following PSF was observed at our hospital. This prompted a review of the data from PSF surgeries performed in our institution between 2019 and 2024 to identify possible risk factors for SSI.

## 2. Materials and Methods

After Institutional Review Board (IRB) approval, we performed a retrospective observational study of the medical records of patients who underwent PSF from 2019 to 2024 in the orthopaedic department of Penteli Children’s Hospital. Forty-six consecutive patients accounting for 67 surgeries performed between 2019 and 2024 were included in the study. Inclusion criteria were as follows: definitive PSF or revision PSF in a patient under 22 years of age for congenital, neuromuscular, or idiopathic spine deformity with at least 9 months post-surgery follow-up for the presence of deep wound infection. Patients treated with PSF for traumatic spinal injuries were excluded from the study.

Patients’ characteristics such as age, sex, Body Mass Index (BMI), and type of deformity were recorded. History of previous spine surgeries, underlying medical illnesses, and nutritional status in terms of total protein and albumin levels were also recorded. Intraoperative factors including duration of surgery, estimated blood loss, class of antibiotics and number of doses administered, instrumentation in sacral vertebrae, and restriction of surgery traffic were also included in the study. Postoperative factors including postsurgical transfer to an Intensive Care Unit (ICU), re-admission to ICU, duration and type of antibiotics administered, and total length of hospitalisation were also recorded.

Before June 2021 all patients undergoing PSF were hospitalised in single-bed rooms. After June 2021 due to a redistribution of patient wards, all patients after PSF were hospitalised in shared hospital wards.

Standard protocol was unchanged throughout the study period. Patients had preoperative nasal cultures for *S. aureus* screening and, if positive, were treated with a mupirocin and chlorhexidine bath for 5 days. They also had a urine culture and, if positive, were treated accordingly. In addition, they underwent a preoperative nutritional assessment in terms of BMI, total protein, and albumin blood levels and, if needed, surgery was postponed until the nutritional status was optimised. The day before surgery, all patients had a chlorhexidine skin wash and a haircut if necessary. Preoperative teicoplanin (10 mg/kg—max 400 mg) and cefuroxime (50 mg/kg—max 1.5 g) were administered 1 h before the surgical incision and continued postoperatively for 48 h. Treatment duration and antibiotic class administered were individualised based on microbiological data and postoperative clinical condition. Intraoperative irrigation was used in all patients, and vancomycin powder was added in the irrigation fluids. A suction drain was placed in the wound before closure and removed when drainage was less than 30 mL/8 h. Postoperatively, patients were admitted to the paediatric Intensive Care Unit if complications were present. Patients with no complications returned to the orthopaedic ward. Electronic and printed medical archives of patients who were included in this study were reviewed for SSI, according to the criteria mentioned above.

Definitions of infection:

A deep SSI was defined, according to CDC criteria, as an infection involving the deep soft tissues of the incision (muscle, fascia, or bone) occurring within 90 days from surgery (with the procedure date counted as day 1), in a patient with purulent drainage from the deep incision requiring surgery or aspiration by a surgeon, having fever (>38 °C), and/or localised pain or tenderness and positive aspiration cultures (Table 1) [6].

We evaluated the rate of SSI after PSF in our institution and possible risk factors including age, sex, length of surgery, total blood loss, type of spinal deformity, admission in the ICU, preoperative BMI, sacral instrumentation, and history of previous spine surgery. We compared the rates of postoperative infection before and after the redistribution of patient wards in mid-2021.

Statistical analysis was performed in SPSS (version 29.0.2.0 (20)). Fisher’s exact test and Pearson’s chi-square analysis were used for categorical variables and Student’s T-test for continuous parametric variables. All statistical analyses were performed as per procedure.

An Institutional Review Board approval was obtained for this study.

## 3. Results

### 3.1. Characteristics of Study Populations: Age, Sex, Scoliosis Subtypes

A total of 46 patients (67 surgeries) were included in the study. The age of patients at the time of surgery ranged from 5 to 22 years (mean: 13.3 years ± 3.6, 95% C.I: 12.4–14.2, median: 13.5 years). Overall, 43 (64.2%) surgeries involved female patients, 34 (50.7%) involved patients with neuromuscular scoliosis, 18 (26.9%) patients with syndromic type, 12 (17.9%) patients with idiopathic scoliosis, and 3 (4.5%) patients with a congenital spinal deformity (Table 2).

### 3.2. Rate of Deep SSI

A total of four cases of deep SSI were recorded. The overall rate of deep SSI in our institution between 2019 and 2024 was 6.0%. Three patients with neuromuscular scoliosis and one patient with syndromic scoliosis presented deep SSI. The rates of deep SSI in the neuromuscular and syndromic cohort were 8.8% and 5.6%, respectively.

### 3.3. Risk Factors for SSI

The mean age of patients from the SSI cohort was higher than the mean age from the non-SSI cohort (17.9 ± 3.3, 95% C.I: 12.6–23.2 vs. 13.0 ± 3.5, 95% C.I: 12.1–13.9, *p* = 0.008). Previous spine surgery was identified as a risk factor for SSI (*p* = 0.031). There was no statistically significant difference between sex, length of surgery, estimated intraoperative blood loss, instrumentation in sacrum, type of deformity, admission to the ICU, and BMI between the two cohorts (Table 3).

### 3.4. Causative Organisms and Treatment

The causative organisms in the cases with deep SSI were *Pseudomonas aeruginosa* in the first, *S. aureus* in the second, *Pseudomonas aeruginosa* and *Enterobacter* in the third, and *E. coli* ESBL in the fourth.

For the first patient a total of 22 days of antibiotic prophylaxis and one operative irrigation and debridement were needed. Maintenance of implants was achieved, and 3 years later there was no significant loss of correction or other complication.

In the second patient, implant prominence led to wound dehiscence. Initial treatment was operative irrigation, debridement, and wound closure, but due to implant prominence and localised skin necrosis in the affected region, the decision was made for a partial implant removal in the affected site. A total of 20 days of antibiotic prophylaxis was administered in this case. Two years later there was no significant loss of correction or other major complication, but surgery for partial revision on the affected site is needed.

In the third case, the deep wound infection progressed to meningitis. Due to the patient’s haemodynamic instability, complete removal of implants was performed. One year later, the patient has no signs of infection, but surgery for PSF is needed in order to correct the deformity.

In the fourth case the initial treatment with antibiotics failed, and the patient needed two consecutive operative irrigation and debridement procedures. A total of 4 months of antibiotic prophylaxis was administered. Implants were maintained, and one year later the patient had no loss of correction or other major complication (Table 4).

### 3.5. Length of Stay

Patients who developed an SSI had a longer length of stay in the hospital than any other patient (*p* < 0.001). The mean length of stay was 34.7 ± 10.6, 95% C.I: 17.8–51.9 days (range: 22 to 44 days) in the SSI cohort and 10.6 ± 6.9, 95% C.I: 8.9–12.4 days (range 2 to 33 days) in the no-SSI cohort.

### 3.6. SSI Before and After Bed Redistribution

Patients were then divided into two groups depending on the time of bed redistribution at the orthopaedic ward: 22 (47.8%) of these patients had spinal fusion before mid-2021, with 28 surgeries (41.8%), whereas 24 (52.2%) had spinal fusion after mid-2021, with 39 surgeries (58.2%). Mean age at time of surgery was 11.0 ± 2.9, 95% C.I: 9.8–12.1 years in the first group and 14.9 ± 3.2, 95% C.I: 13.9–16.0 years in the second group (*p* < 0.01). There was no other statistically significant difference between the two groups in variables such as sex, BMI, history of previous surgery, total intraoperative blood loss, instrumentation in sacrum, type of deformity, or need for admission in the ICU. Moreover, despite the fact that the main spinal surgeons remained the same in both periods, the mean total length of surgery was higher in the group after mid-2021 than that before the mid-2021 period (400.2 ± 119.9, 95% C.I: 361.4–439.1 min vs. 319.3 ± 131.2, 95% C.I: 268.4–370.2 min, *p* = 0.011) (Table 5). Finally, preoperative, intraoperative, and postoperative protocols were the same in both periods. Patients treated after mid-2021 were hospitalised in a shared room as opposed to patients hospitalised before mid-2021 who were staying in single-bed rooms. We identified all four cases of deep SSI in our PSF cohort presenting in the late period, after mid-2021; however, there was no statistically significant difference in developing SSI between the two groups.

## 4. Discussion

### 4.1. Main Findings

Despite huge efforts, SSIs remain a severe complication after PSF in paediatric patients.

In our institution the overall rate of deep SSI was 6.0%. Taking into consideration the fact that our study included many high-risk patients, this rate is among the expected rates [7,8]. History of prior surgery (*p* = 0.031) and higher age at time of surgery (*p* < 0.001) were identified as risk factors for developing an SSI. There was no difference between the mean age at time of surgery in patients who had a history of previous spine surgery and those who did not (13.4 ± 4.1, 95% C.I: 11.8–14.9 vs. 13.2 ± 3.3, 95% C.I: 12.1–14.3) (*p* = 0.853). Many previous studies have tried to identify perioperative risk factors and patient characteristics that are associated with a high risk of developing an SSI postoperatively [5,9,10,11].

### 4.2. Comparison with the Prior Literature

A study of Rudik TN. et al. in 9801 patients who underwent primary fusion for AIS, reported that obesity and male sex were significantly associated with postoperative infection. In that study, length of fusion was not a significant factor in developing an SSI. The 90-day incidence of SSI in that study was 2.7% [5].

Another recent study in 4145 patients who underwent PSF for NMS and/or cerebral palsy between 2015 and 2020, with data from the American College of Surgeons National Surgical Quality Improvement Program (NSQIP), reported a 2.5% prevalence of deep SSI and identified American Society of Anesthesiologists (ASA) classification ≥ 3, increased BMI, preoperative corticosteroids, preoperative osteotomy, prolonged anaesthesia, prolonged operative time, and postoperative UTI as risk factors [9].

In another study from a single tertiary care children’s hospital the reported SSI rate was 4.4% after paediatric PSF. Risk factors included obesity, antibiotic prophylaxis with clindamycin, inappropriately low dose of antibiotics, longer duration of hypothermia, and ASA score greater than 2. In agreement with the results of our own study, no scientifically significant association between SSI and duration of surgery and fusion of sacral vertebra was found [10].

Other recent studies reported SSI rates between 4.2% and 6.3%, which are close to the rates observed in our own study. Neuromuscular scoliosis, previous spinal surgery, sacral vertebrae fusion, use of allograft, no use of drain, blood transfusion, obesity, amount of intraoperative crystalloids, ventriculoperitoneal shunt, non-ambulatory status, pelvic instrumentation, procedure time ≥ 7 h, American Society of Anesthesiologists grade > 2, revision procedure, hospital spine surgical cases < 100/year, and abnormal haemoglobin levels were recognised as risk factors [8,12,13,14,15,16,17,18,19].

However, other studies failed to prove an association between previous spine surgery, length of surgery, and SSI [20,21].

A retrospective review from Glotzbecker et al. in 2013 concluded that there was fair evidence that underlying medical condition, neuromuscular disease, urinary or bowel incontinence, inappropriate perioperative antibiotic prophylaxis, and implant prominence were associated with a higher risk of SSI after PSF, while there was conflicting evidence regarding positive urine cultures, obesity, pre- or postoperative malnutrition, blood loss, blood transfusions, number of levels fused, extension of fusion to the sacrum, prolonged operative time, no use of drains, and increase in SSI rates [18].

The 2022 Best-Practice Guidelines (BPGs) for SSI in high-risk paediatric spine surgery reported a syndromic and neuromuscular aetiology of scoliosis, BMI < 15 or >30, preoperative albumin < 3 mg/dL, ASA score IV or higher, diabetes, preoperative steroid use, non-ambulatory status, presence of baclofen pump, presence of ventriculoperitoneal shunt, urinary incontinence, positive urine cultures, thirteen levels of fusion or greater, and fusion to pelvis as risk factors for developing an SSI [7].

In three (75%) of the cases with SSI in our institution, one microorganism was isolated while in the fourth, two microorganisms were isolated. The microorganisms were *P. aeruginosa* in 2/4 (50%) of cases, *S. aureus* in 1/4 (25%), *E. coli* ESBL in 1/4 (25%), and *Enterobacter* in 1/4 (25%). For every case all the efforts were directed towards retaining the implants, in accordance with the evidence published in the literature [22]. Although retaining implants is associated with a longer duration of antibiotic prophylaxis, removing implants and risking the correction of deformity already achieved can be disastrous [23]. Of our cases, two (50%) maintained the implants while the other two eventually needed partial or total implant removal.

### 4.3. Possible Mechanisms

It must be noted that many of the microorganisms associated with SSIs tend to form a microbial biofilm. In fact, according to some reports, as many as 80% of SSIs may be related to the formation of a microbial biofilm [24,25]. The biofilm is essentially composed of water, microbial cells, and Extracellular Polymeric Substances (EPSs), including polysaccharides, proteins, lipids, extracellular enzymes, metal ions, and nucleic acids such as extracellular DNA. These constituents provide structural support to the biofilm and protect the microorganisms from antimicrobials and neutrophils [26].

As most antimicrobials are developed and tested on planktonic bacteria, they are ineffective against biofilms. The biofilm not only poses a physical barrier against antimicrobials and neutrophils but also has unique characteristics that make it resistant to antibiotics. One of them, Quorum Sensing (QS), facilitates the transfer of resistant genes between neighbouring bacteria. Besides QS, the depletion of nutrients and oxygen inside the biofilm lead to slower cell growth rate, which makes bacteria resistant to some antibiotics [26].

All four cases of deep SSI appeared in patients who underwent PSF after mid-2021 and were hospitalised in shared rooms. Despite the fact that this observation was not statistically significant, it is something that has to be taken into consideration. Further studies of the postoperative conditions and nursing care of patients who underwent PSF with larger samples are needed.

No case of superficial SSI was observed in this study. This observation may be due to the fact that, in our department, patients with a superficial SSI are treated on an outpatient basis. Another factor that may contribute to this observation is the fact that superficial SSI causes mild symptoms and produces faint signs of infection, so it can be easily misdiagnosed.

### 4.4. Limitations

Limitations of this study include the retrospective design, the single-centre scope, the small number of patients included, and the lack of adjustment for confounders, which limits the statistical power. Another limitation is the great variety of spinal deformity between the patients included in the study. All these limitations make it difficult to draw statistically significant conclusions that could be applied to the entire population.

## 5. Conclusions

Despite significant efforts, SSI remains a major complication after PSF for spinal deformities in children and adolescents. Between 2019 and 2024 four cases of deep SSI occurred in our institution, leading to a rate of 6.0% among all PSF for this specific time period. Higher age and a history of previous spine surgery were risk factors for SSI in this cohort of patients. All cases of deep SSI occurred in the time period after mid-2021, when a redistribution of patient wards took place resulting in patients being hospitalised in shared rooms instead of single-bed rooms. While this finding may be attributable to chance, it is an observation that needs further investigation.

## Figures and Tables

**Table 1 healthcare-13-03043-t001:** Deep SSI CDC criteria.

Deep Incisional SSI As Per CDC Criteria
1. Occurs within 90 days following the operative procedure (where day 1 = the procedure date) AND;
2. Involves deep soft tissues of the incision (muscle, fascia) AND;
3. Patient has at least one of the following: • Purulent drainage from the deep incision; • A deep incision that is deliberately opened or aspirated by a surgeon or spontaneously dehisces AND; ✓ Organism(s) identified from the deep soft tissues of the incision by a culture AND; ✓ Patient has fever (>38 °C) and/or localised pain or tenderness; • An abscess or other evidence of infection involving the deep incision detected on gross anatomical exam, histopathologic exam, or imaging test.

**Table 2 healthcare-13-03043-t002:** Patient characteristics.

Variable	Total Procedures (*n* = 67)
Age at procedure, mean (range), years	13.3 (5.0–22.0)
Female sex	43 (64.2%)
Male sex	24 (35.8%)
Diagnosis:	
Idiopathic spinal deformity	12 (17.9%)
Neuromuscular	34 (50.7%)
Syndromic	18 (26.9%)
Congenital	3 (4.5%)

**Table 3 healthcare-13-03043-t003:** SSI vs. no SSI risk factors.

Variable	No SSI (*n* = 63)	SSI (*n* = 4)	*p*-Value
Age (mean)	13.0	17.9	0.008
Sex:			0.542
Female	41	2	
Male	22	2	
BMI (mean)	18.4	17.2	0.540
Length of surgery (mean, min)	370	310	0.375
Previous spine surgery	25	4	0.031
Intraoperative blood loss (mean, mL)	1040	782.5	0.567
Sacrum instrumentation	27	1	0.635
Type of deformity:			0.694
Idiopathic	12	0	
Neuromuscular	31	3	
Syndromic	17	1	
Congenital	3	0	
Admission in the ICU	52	4	0.361

**Table 4 healthcare-13-03043-t004:** Patients with SSI.

	Organism	Days of Antibiotic Treatment	Operative Irrigation and Debridement	Retention of Implants
Patient 1	*P. aeruginosa*	22	1	Yes
Patient 2	*S. aureus*	20	1	Partial
Patient 3	*P. aeruginosa* *Enterobacter* spp.	45	0	No
Patient 4	*E. coli* ESBL	120	2	Yes

**Table 5 healthcare-13-03043-t005:** Cohort characteristics before vs. after the redistribution of wards.

VARIABLE	BEFORE Mid-2021 (*n* = 28)	AFTER Mid-2021 (*n* = 39)	*p*-Value
Age (mean, years)	11	14.9	<0.001
Sex:			0.617
Female	19	24	
Male	9	15	
BMI (mean)	19.2	17.7	0.133
Length of surgery (mean, min)	319.3	400.3	0.011
Previous spine surgery	12	17	0.952
Intraoperative blood loss (mean, m)	827.1	1166.4	0.113
Sacrum instrumentation	8	20	0.081
Type of deformity:			0.068
Idiopathic	6	6	
Neuromuscular	9	25	
Syndromic	11	7	
Congenital	2	1	
Admission in the ICU	21	35	0.180
SSI	0	4	0.134

## Data Availability

The original contributions presented in this study are included in the article. Further inquiries can be directed to the corresponding author(s).

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
