# Peer review of "Surgical Site Infection After Posterior Spinal Fusion for Paediatric Spinal Deformities: A Single-Centre Retrospective Observational Study"

_healthcare, 2025, doi:10.3390/healthcare13233043_

Round 1

Reviewer 1 Report

Comments and Suggestions for Authors

Dear authors,

Thank you for the opportunity to review your work on this important topic. Your efforts to systematically collect data on SSI in this vulnerable patient population are appreciated, and the manuscript has the potential to be informative if several key points are clarified and tempered. I encourage you to more clearly define the primary aim and novelty of the study, explicitly acknowledge the limitations of the very small number of SSI events, and avoid strong terms such as “independent risk factors” or causal implications regarding ward redistribution without appropriate multivariable analysis. Clarifying whether analyses are per patient or per procedure, using statistical tests suitable for small cell counts (e.g., Fisher’s exact test), providing effect sizes with confidence intervals, and ensuring that all tables, p-values, and abbreviations are consistent will strengthen the scientific rigor. It would also be helpful to discuss more transparently the likely under-detection of superficial SSI, refine the description of microbiological findings and antibiotic management with accurate terminology, and streamline the discussion to focus on how your descriptive data fit within existing literature. A careful language edit will further improve clarity and readability.

Comments on the Quality of English Language

 The English could be improved to more clearly express the research.

Author Response

Thank you very much for taking the time to review this manuscript. Please find the detailed responses below and the corresponding revisions/corrections highlighted in the re-submitted files. 

Comment 1: Define more clearly  the primary aim and novelty of the study.

Response 1: Thank you for pointing this out. We added the term '' the overall rate of SSI'' in abstract line 18 in order to more clearly define the primary aim of the study.

Comment 2: Explicitly acknowledge the limitations of the very small number of SSI events, and avoid strong terms such as “independent risk factors” or causal implications regarding ward redistribution without appropriate multivariable analysis.

Response 2: Thank you for pointing this out. We agree with this comment therefore we added in the limitations paragraph 4.4 the term ''the lack of adjustment for confounders, which limits the statistical power'' in order to highlight the limitations that arise from the small number of SSI and from the lack of multivariate analysis. We also erased the term ''independent'' throughout the manuscript. We reformed the last sentence in pragraph 5.conclusions and added the sentence ''While this finding may be attributable to chance'' in order to avoid causal implications between redistribution of wards and SSI.

Comment 3: Clarifying whether analyses are per patient or per procedure, using statistical tests suitable for small cell counts (e.g., Fisher’s exact test), providing effect sizes with confidence intervals, and ensuring that all tables, p-values, and abbreviations are consistent will strengthen the scientific rigor.

Response 3: Thank you for pointing this out. We agree with this comment therefore in paragraph 2.Materials and Methods we added the sentence ''All statistical analysis was performed per procedure'' in order to clarify this point. We performed statistical analysis using Fisher's extract test were appropriate and we corrected the p-values accordingly. We also clarified in the paragraph 2.Materials and Methods line 110-111 that apart from Pearson’s Chi-square analysis we also used Fisher's extract test. We added effect sizes with Confidence Intervals wherever appropriate throughout the text and we re-examined and corrected tables, p-values and abbreviations in order to be consistent.

Comment 4: It would also be helpful to discuss more transparently the likely under-detection of superficial SSI, refine the description of microbiological findings and antibiotic management with accurate terminology, and streamline the discussion to focus on how your descriptive data fit within existing literature.

Response 4: Thank you for pointing this out. We tried to be as transparent as possible regarding the likely under-detection of superficial SSIs. The fact that they present with subtle clinical signs, combined with the subjective nature of the diagnosis, makes it difficult to accurately identify affected patients. Regarding the description of microbiological findings and antibiotic management we aimed to concisely present the most essential points for this study, including the microorganisms isolated in patients with SSI, the overall treatment administered to patients with SSI, as well as the specific prophylactic drugs and dosages used in all surgeries. We believe that we have sufficiently analyzed the findings of our study in relation to the existing literature and have also incorporated several recent studies (2023–2025).

Comments on the Quality of English Language: Regarding the comments on language, we are willing to use the professional language editing service in the last edition of our manuscript, after we include all your corrections. 

Reviewer 2 Report

Comments and Suggestions for Authors

The manuscript addresses an important and clinically relevant topic: surgical site infections (SSIs) following posterior spinal fusion for pediatric spinal deformities. The study is timely, well-motivated, and generally well-organized, but several areas require clarification and refinement to strengthen methodological rigor and interpretability.

Comments

  1. The manuscript should explicitly define the study as a retrospective observational study and maintain this terminology consistently throughout the title, abstract and main text. The current wording may cause confusion between retrospective and prospective elements.

  2. With only four infection cases, statistical power is limited. The authors should explicitly acknowledge this constraint in the Discussion and moderate the strength of causal statements.

  3. The results rely on univariate comparisons. If possible, a multivariate logistic regression (including age, sex, BMI, diagnosis type and prior surgery) should be conducted to confirm independence of identified risk factors. Otherwise, justify why multivariate analysis was not feasible.

  4. Results presentation. Include measures of dispersion (mean ± SD) and 95% confidence intervals where appropriate, clarify whether p-values were adjusted for multiple testing and ensure consistency in decimal places and units across tables.

  5. The finding that all SSI cases occurred after mid-2021 is intriguing. Please expand this discussion (were there differences in room ventilation, staffing ratios, postoperative care, or infection control policies that could explain this pattern?).

  6. The discussion is informative but dense. Consider reorganizing it into clearly defined subsections: (a) main findings, (b) comparison with prior literature, (c) possible mechanisms and (d) limitations.

  7. Explicitly list the study’s limitations, including small sample size, retrospective design, single-center scope, heterogeneity in patient diagnoses and lack of adjustment for confounders.

  8. The reference list is comprehensive but could benefit from adding recent studies or meta-analyses (2023–2025) addressing SSI prevention or risk modeling in pediatric spinal surgery. Ensure all citations follow MDPI formatting standards and include valid DOIs.

  9. Ensure that all tables and figures include titles, legends and consistent formatting.

Comments on the Quality of English Language

The English language is generally clear and understandable, but several sentences could be restructured to improve readability and flow. Minor grammatical corrections, punctuation adjustments and consistent use of medical terminology are recommended.

Specifically:

  • Simplify long sentences to enhance clarity.

  • Ensure consistent use of tense.

  • Unify terminology.

  • Review prepositions and articles for smoother phrasing.

A careful language edit by a fluent English speaker or professional editing service would improve the overall polish of the manuscript.

Author Response

Thank you very much for taking the time to review this manuscript. Please find the detailed responses below and the corresponding revisions/corrections highlighted in the re-submitted files. 

Comments 1: The manuscript should explicitly define the study as a retrospective observational study and maintain this terminology consistently throughout the title, abstract and main text. The current wording may cause confusion between retrospective and prospective elements.

Response 1: Thank you for pointing this out. We agree with this comment. Therefore we included the term '' A Single-Center Retrospective Observational Study '' in the title, we added the term '' retrospective observational '' in the Abstract, line 21 and the term ''observational'' in the paragraph 2.Materials and Methods, page 2, line 57. 

Comment 2: With only four infection cases, statistical power is limited. The authors should explicitly acknowledge this constraint in the Discussion and moderate the strength of causal statements.

Response 2: Thank you for pointing this out. We added the sentence "which limits the statistical power" in the discussion section, paragraph 4.4, line 283. We also erased the term ''independent'' from Abstract line 28, from paragraph 3.3 line 134, from paragraph 4.1 line 198 and from paragraph 5 line 292.

Comments 3: The results rely on univariate comparisons. If possible, a multivariate logistic regression (including age, sex, BMI, diagnosis type and prior surgery) should be conducted to confirm independence of identified risk factors. Otherwise, justify why multivariate analysis was not feasible.

Response 3: Thank you for pointing this out. We have conducted a multivariate logistic regression which could not confirm the independence of any risk factor. In fact the standard errors and odds ratios for most of the variables were extreme, possibly due to small sample size. We consider that adding this analysis to the manuscript would not be constructive at this stage. However, we will include in the limitations section that the results have not been confirmed through a multivariate analysis.

Comments 4: Results presentation. Include measures of dispersion (mean ± SD) and 95% confidence intervals where appropriate, clarify whether p-values were adjusted for multiple testing and ensure consistency in decimal places and units across tables.

Response 4: Thank you for pointing this out. We included measures of dispersion and 95% confidence intervals in several points throughout the text. Paragraph 3.1 line 118, Paragraph 3.3 line 133, paragraph 3.5 line 166 and 167, paragraph 3.6 line 173- 174, Paragraph 3.6 line 180-181, paragraph 4.1 line 200-201. We have thoroughly re-examined the manuscript and corrected decimal places in order to be consistent.

Comments 5: The finding that all SSI cases occurred after mid-2021 is intriguing. Please expand this discussion (were there differences in room ventilation, staffing ratios, postoperative care, or infection control policies that could explain this pattern?). 

Response 5: The primary difference between the periods before and after mid-2021 was that, following mid-2021, all patients were transferred to shared hospital rooms postoperatively, whereas prior to mid-2021, patients were accommodated in single-bed rooms after surgery.

Comments 6: The discussion is informative but dense. Consider reorganizing it into clearly defined subsections: (a) main findings, (b) comparison with prior literature, (c) possible mechanisms and (d) limitations.

Response 6: Thank you for pointing this out. We agree with this comment. Therefore we reorganised the Discussion section and added 4 subsections according to your recommendations. 

Comments 7: Explicitly list the study’s limitations, including small sample size, retrospective design, single-center scope, heterogeneity in patient diagnoses and lack of adjustment for confounders.

Thank you for pointing this out. We agree with this comment. Therefore we added '' single- center scope'' and ''lack of adjustment for confounders'' in the limitations of the study, page 8, paragraph 4.4, line 282. 

Comments 8: The reference list is comprehensive but could benefit from adding recent studies or meta-analyses (2023–2025) addressing SSI prevention or risk modeling in pediatric spinal surgery. Ensure all citations follow MDPI formatting standards and include valid DOIs.

Response 8: Thank you for pointing this out. We revised the references and formatted them to comply with MDPI standards. Unfortunately there are not many recent studies or meta-analyses addressing SSI prevention or risk modeling in pediatric spinal surgery published in the literature. We believe that we have included all recent studies (2023-2025) which are relevant to our study.

Comment 9: Ensure that all tables and figures include titles, legends and consistent formatting.

Response 9: Thank you for pointing this out. We revised the figures and tables in order to have consistent formatting. 

Response on comments on the Quality of English Language: Thank you for your comments. We aim to ensure that the manuscript is clear and that the data are presented in the most appropriate way. If you believe that using a professional editing service would help convey the information more effectively, we will proceed with it once we have the final version of the manuscript, after incorporating your revisions.